# The Role of Psychological Flexibility and Psychological Factors in Chronic Pelvic Pain Among Women: A Correlational Study

**DOI:** 10.3390/healthcare13141697

**Published:** 2025-07-15

**Authors:** Chiara Manna, Michelle Semonella, Giada Pietrabissa, Gianluca Castelnuovo

**Affiliations:** 1Department of Psychology, Catholic University of Milan, 20123 Milan, Italy; michelle.semonella@biu.ac.il (M.S.); giada.pietrabissa@unicatt.it (G.P.); gianluca.castelnuovo@unicatt.it (G.C.); 2Department of Psychology, Bar-Ilan University, Ramat Gan 590002, Israel; 3IRCCS Istituto Auxologico Italiano, Clinical Psychology Research Laboratory, San Giuseppe Hospital, 28824 Verbania, Italy

**Keywords:** chronic pelvic pain, Psychological Flexibility, correlations

## Abstract

**Background/Objectives:** Chronic Pelvic Pain (CPP) is a multifactorial condition that affects in many ways the daily life of patients suffering from it. Different psychological factors demonstrated to be associated with the genesis and maintenance of CPP. Less is known about the role of the Psychological Flexibility (PF) model. Thus, the aim of this study is to explore the relationship between the PF domains, psychological distress, pain, and quality of life in patients with chronic pelvic pain. **Methods:** A total of 114 women with a diagnosis of chronic pelvic pain were included in this study. Participants completed online self-report measures to assess psychological distress (anxiety, depression, stress), Psychological Flexibility, Pain interference, and Quality of life. **Results:** Psychological distress and Psychological Flexibility showed significant association with pain interference. Other PF dimensions related to pain interference were as follows: self as context, defusion, and values. Physical Quality of life showed significant association with Experiential avoidance and Lack of values clarity, while Mental Quality of life was associated with Psychological Inflexibility and Self as content. **Conclusions:** Psychological distress and Psychological Flexibility have a role in pain perception and its interference with a patient’s daily life, affecting also physical and mental quality of life of CPP patients.

## 1. Introduction

Chronic pelvic pain (CPP) is defined as pain lasting over 6 months in the pelvic region which cannot be explained by infection or other obvious local pathology [1]. CPP is estimated to affect about 15% of the general population [2], is often associated with negative emotional, cognitive, behavioural, and sexual consequences and is a cause of disability, poor health-related quality of life, and increased healthcare and medication utilization [3]. CPP is an ‘umbrella term’ that encompasses various pathologies of urologic and gynecological origin (e.g., endometriosis, vulvodynia, vaginismus, chronic prostatitis, interstitial cystitis, and bladder pain syndrome). This broad classification can thus create confusion among patients, physicians, and within the scientific literature [4,5]. Chronic pelvic pain (CPP) encompasses conditions of various origins, including gynecological, urological, and musculoskeletal, which may arise from inflammatory or, in some cases, malignant processes [6,7]. Diagnosis typically involves a combination of symptom assessment and physical examination. However, due to the diagnostic heterogeneity, the use of imagining instruments (e.g., ultrasound, sonography, computed tomography) can be useful to discriminate between CPP diagnosis. Transvaginal ultrasonography (TVUS) is considered the first-line screening tool for CPP, providing initial insights into possible gynecological causes. Magnetic resonance imaging (MRI) of the pelvis offers greater specificity and is often used to further investigate findings from ultrasonography or to evaluate potential malignancies [8]. Additionally, B-mode ultrasound can be used to examine structural and morphological abnormalities of the uterus, while Doppler ultrasound is particularly useful for detecting gynecological tumours by assessing blood flow and vascular patterns [9]. CPP symptoms arise from an interplay between physical processes, such as tissue inflammation, dysfunctions in the immune, endocrine or nervous systems, and psychological factors [3,10]. Psychological factors play a dual role in CPP. Patients with CPP tend to report higher rates of psychiatric comorbidities [11]—particularly anxiety, depression, personality disorders, and PTSD [12,13,14,15]. While the association between these psychological variables and chronic pain, as well as their impact on patients’ quality of life, is well-documented in the literature [3,10], it remains unclear whether psychological distress is a cause or a consequence of CPP. Evidence suggests an interaction between these factors, generating a vicious cycle that exacerbates pain and diminishes both physical and mental quality of life for affected individuals [16].

Recent studies highlighted the role of psychological factors in the genesis and maintenance of different types of pelvic pain. As in other chronic pain conditions, depression, anxiety, and pain catastrophizing are the most common associated factors in CPP [15,17,18]. Among the psychological factors, there is a growing interest about the role of Psychological Flexibility (PF) in the field of chronic pain [19]. Psychological Flexibility is the core concept of Acceptance and Commitment Therapy (ACT) and it is defined as “the ability to contact the present moment more fully and to change or persist in behaviour when doing so serves valued ends” [20]. It is composed of 6 core processes: acceptance, defusion, contact with the present moment, self as a context, values, and committed action which together form the “hexaflex model”. The opposite of Psychological Flexibility is Psychological Inflexibility (PI), which is characterized by six opposing processes: experiential avoidance, cognitive fusion, lack of contact with the present moment, self as content, lack of alignment with values, and inaction. The hexaflex model can also be simplified into the triflex model, which groups the six processes into three categories: openness (including acceptance and defusion), awareness (including self as context and contact with the present moment), and engagement (including values and committed action) [21].

Evidence considers PF as a mental health protective factor: it is associated with adaptive behaviours and response to distress and better mental health outcomes [22,23]. On the other hand, lower levels of PF (or Psychological Inflexibility) are related to more dysfunctional behaviours, psychopathology, and poor mental health [24,25]. The Psychological Flexibility model is gaining increasing attention for the management of various chronic pain conditions [26,27]. However, research on the role of Psychological Flexibility (PF) in chronic pelvic pain is still scarce [28]. To date, most studies have focused on specific CPP conditions or individual components of the hexaflex model, yielding some promising preliminary findings [28,29,30]. Given these insights, the PF model could be a valid ally in the integrated management of CPP, as recommended by current guidelines [1]. Thus, a deep knowledge of its involvement in the genesis and maintenance of CPP is needed.

Therefore, the purpose of this study is to explore the association of psychological distress, Psychological Flexibility, Psychological Inflexibility and their subcomponents, pain, and quality of life of women with chronic pelvic pain.

## 2. Materials and Methods

This study followed the guidelines from the STROBE checklist for cohort studies [31]. This was a correlational study of women with chronic pelvic pain. Data were collected using the online survey platform Qualtrics between September 2023 and June 2024. Participants were eligible if they met the following inclusion criteria: being over 18 years of age, having a documented diagnosis of chronic pelvic pain, and fluently speaking and understanding Italian. Exclusion criteria included a diagnosis of severe depression, personality disorders, or other documented psychiatric comorbidities, as well as cognitive impairments that could hinder the ability to provide informed consent. All procedures followed the ethical standards of the Declaration of Helsinki (1964) and its subsequent amendments. The study protocol was approved by the Ethical Committee of the Catholic University of Milan (protocol number: 66-23).

### 2.1. Procedure

The study was advertised in two specialized private gynecological clinics in Milan with the support of healthcare professionals, and on the websites and social media of associations active in the field of chronic pelvic pain. Participants completed self-report questionnaires assessing psychosocial factors and pain outcomes, including measures of anxiety, depression, stress, Psychological Flexibility, and pain interference. Demographic data, duration of pain, and time since diagnosis were also collected.

### 2.2. Measurement Instruments

Psychological distress (i.e., anxiety, depression, stress) and Psychological Flexibility components (acceptance, defusion, contact with the present moment, self as a context, values and committed action) were considered as predictors in this study. Outcome measures were the following: pain interference, and physical and mental quality of life.

The following measurement instruments will be used:Depression Anxiety Stress Scale-21 (DASS-21) [32] was used to assess psychological distress. This scale comprises 21 items divided into three subscales, each measuring anxiety, depression, and stress. Each item is rated on a 4-point Likert scale, ranging from 0 (‘did not apply to me at all’) to 3 (‘applied to me very much or most of the time’). The cut-off scores are as follows: a score of ≥5 for the depression subscale, ≥4 for the anxiety subscale, and ≥8 for the stress subscale. We used the Italian validated version of the DASS-21, whose psychometric properties have shown good internal consistency and construct validity in Italian samples [32]. In our sample, internal consistency was adequate for all three subscales: anxiety (Cronbach’s α = 0.802), depression (Cronbach’s α = 0.876), and stress (Cronbach’s α = 0.842)Multidimensional Psychological Flexibility Inventory (MPFI) [33] was used to assess Psychological Flexibility and its subcomponents. It consists of 60 items, each rated on a 6-point Likert scale ranging from 1 (‘never true’) to 6 (‘always true’). This instrument comprises two subscales: one measuring Psychological Flexibility (30 items) and the other measuring Psychological Inflexibility (30 items). The Italian validated version of the MPFI was used [33]. Internal consistency for our sample was adequate for both subscales: Psychological Flexibility (Cronbach’s α = 0.950) and Psychological Inflexibility (Cronbach’s α = 0.942).Brief Pain Inventory (BPI) [34,35] was used to assess both pain intensity and pain interference. This instrument consists of two subscales: one evaluating pain intensity and the other evaluating pain interference with daily activities (walking, mood, sleep, work, relationships, and enjoyment of life). The BPI includes a total of 15 items rated on a Numeric Rating Scale (NRS) from 0 to 10. Additionally, a separate NRS ranging from 0 to 10 is used to assess the patient’s current pain intensity. Scores can be calculated for pain intensity, pain interference, or as a total score, with higher scores indicating greater pain severity and interference. We used the Italian validated version of the BPI [35], which showed good internal consistency and construct validity in Italian samples. In our sample, internal consistency was adequate both for pain interference (Cronbach’s α = 0.863) and pain intensity (Cronbach’s α = 0.815).Short Form-12 Health Survey (SF-12) [36] was used to assess both physical and mental quality of life. This instrument is a shorter version of the SF-36 Health Survey and includes 12 items that assess eight dimensions of an individual’s life that may be affected by disease. Responses are given using either dichotomous yes/no answers or 3- or 5-point Likert scales. Higher scores reflect better quality of life. We used the Italian validated version of the SF-12, whose psychometric properties have shown good internal consistency and construct validity in Italian samples [37].

### 2.3. Power Calculations

An a priori sample size estimation was performed using regression analysis through G*Power software (version 3.1). With a regression model including ten predictors and assuming a medium effect size (f^2^ = 0.15) for each independent variable, and a power level of 0.80, a sample size of at least 119 participants would be needed.

### 2.4. Statistical Analysis

Statistical analyses were run using Jamovi software (version 2.3.28 for Windows) [38]. Continuous variables were expressed as mean (M) and standard deviation (SD), while categorical variables were presented as frequencies and percentages. The normality of the data distribution was assessed using the Shapiro–Wilk test, along with evaluations of skewness and kurtosis.

Due to the inability to achieve the required statistical power given the limited sample size, it was not possible to perform traditional parametric correlations. As a result, we opted for non-parametric statistical analyses to assess the associations between psychological variables, pain, and quality of life. Specifically, Spearman’s rank-order correlation (Spearman’s rho) was employed, with a *p*-value < 0.05 considered indicative of statistical significance. Correlation coefficients (rho) were interpreted as follows: weak (0–0.3), moderate (0.3–0.5), strong (0.5–0.7), and very strong (0.7–1).

## 3. Results

A total of 114 Italian women were included in this study. The mean age of participants was 35.1 years old (SD: 20.0). All participants were diagnosed with mixed comorbidities related to chronic pelvic pain. The most common primary diagnosis was endometriosis (34.8%), followed by vulvodynia (30.4%), and general chronic pelvic pain (6.1%). Other diagnoses were the following: vestibulodynia, adenomyosis, interstitial cystitis hypertonic pelvic floor dysfunction, pudendal neuralgia, or other pelvic neuropathies. A total of 54.8% of participants resided in Northern Italy, 23.5% in Central Italy, 13.9% in Southern Italy, and 7.0% on the Italian islands. Additionally, 48.7% had completed a university degree. The majority of participants were single (47.8%), while 21.7% were married, and another 21.7% were living with a partner. Finally, 92.2% reported to be suffering from chronic pelvic pain from more than one year.

Descriptive Statistics are shown in Appendix A.

The demographic characteristics of the sample are reported in Appendix A.

### 3.1. Pain Interference, Pain Intensity, and NRS

Spearman Correlations revealed a significant moderate association between anxiety and pain interference (r = 0.324); a moderate association between stress and pain interference (r = 0.481); and a weak association between depression and pain interference (r = 0.302). A moderate association was found between stress and the current pain intensity score (NRS) (r = 0.321). No associations were found between psychological distress and pain intensity. Correlational analysis between anxiety, depression, stress, pain interference, and pain intensity are reported in Appendix A.

Correlations between the Psychological Flexibility domains subscale revealed significant associations between pain interference and the Psychological Flexibility composite score (r = −0.299), self as context (r = −0.302), defusion (r = −0.361), and values (r = −0.257). No associations were found between Psychological Flexibility domains, pain intensity, and the current pain intensity score (NRS).

Correlations between Psychological Flexibility subscales and pain are shown in Appendix A.

Regarding the Psychological Inflexibility subscales, significant associations were found between the current pain intensity score (NRS), Cognitive Fusion (r = 0.332), and Lack of values clarity (r = 0.281). No correlations were found between Psychological Inflexibility domains and pain interference or the composite score of pain intensity.

Correlations between Psychological Inflexibility subscales and pain are shown in Appendix A.

### 3.2. Quality of Life

Significant correlations were found between the mental quality of life domain and the composite score of Psychological Inflexibility (r = 0.239). A moderate association was also found between mental quality of life and Self as content (r = 0.332). No other correlations were found between Psychological Flexibility domains and mental quality of life.

Correlations between Psychological Inflexibility domains and Mental quality of life are shown in Appendix A.

Physical quality of life was significantly associated with Experiential avoidance (r = 0.273) and Lack of values clarity (r = −0.236). No other correlations were found between Psychological Flexibility domains and physical quality of life. No correlations were found between anxiety, depression, stress, and physical or mental quality of life.

Correlations between Psychological Inflexibility and physical quality of life are shown in Appendix A.

### 3.3. Psychological Distress

Spearman’s correlations revealed significant association with the composite score of Psychological Inflexibility and anxiety (r = 0.396); Psychological Inflexibility and depression (r = 0.579); and Psychological Inflexibility and stress (r = 0.470). Almost all of the Psychological Inflexibility subscales showed significant associations with depression and stress scores.

These correlations are shown in Appendix A.

Correlations between Psychological Flexibility and psychological distress showed significant association between Self as contest (r = −0.273) and stress; and between defusion (r = −0.297) and stress.

Correlations between Psychological Flexibility domains and psychological distress are shown in Appendix A.

## 4. Discussion

The aim of this study was to explore the role of psychological factors, (in particular Psychological Flexibility (PF), Psychological Inflexibility (PI), and their subcomponents) along with pain interference, pain intensity and quality of life in patients with chronic pelvic pain (CPP). To our knowledge, this is the first study exploring the relationship between Psychological Flexibility processes and pain among different CPP conditions.

As expected, our results show that psychological distress (e.g., anxiety, depression, and stress) is associated with pain interference, suggesting that these symptoms intensify patients’ perception of pain. Notably, stress demonstrated a stronger relationship with the current pain intensity (NRS), indicating that higher stress levels are linked to increased pain intensity. This finding is not surprising, as it is well established that acute stress exacerbates pain symptoms, also in chronic pelvic pain [39,40]. The relationship between psychological distress and chronic pain was already reported elsewhere, both in other chronic pain conditions, and in chronic pelvic pain specifically [15,18,41,42]. Nevertheless, the small sample size and the correlational nature of this study limit the ability to draw conclusions about the causality of this relationship. It is therefore possible that pain and psychological distress interact in a cyclical manner, where pain restricts the patient’s daily activities, which in turn worsens their mental distress [10,43].

The total Psychological Flexibility score showed a negative association with pain interference, suggesting that lower flexibility is linked to greater pain dominance in patients’ lives. While the predictive role of Psychological Flexibility in chronic pelvic pain has been documented [28,44], evidence on the specific PF processes involved remains limited.

Previous studies [28,45] have examined the role of pain acceptance in patients with vulvodynia or endometriosis. However, these studies focused on pain acceptance, which is conceptualized differently from acceptance within the Psychological Flexibility (PF) model. As a result, they utilized pain-specific acceptance measures, such as the Chronic Pain Acceptance Questionnaire (CPAQ-8) [46]. Thus, the interpretation of these results appears hampered, and it cannot allow to make comparisons with the construct of acceptance, as conceptualized in the PF model.

In our study, the PF subcomponents that showed a significant association with pain interference were self as context, defusion, and values. Self as context is related to the dimension of acceptance and refers to the ability to observe one’s thoughts, emotions, and experiences from a detached perspective. This skill allows individuals to observe their thoughts and feelings without becoming identified or fused with them [21]. In the Psychological Inflexibility subscale, the opposite of self as context is self as content. This refers to the tendency of individuals to identify with their own thoughts, feelings, and fears. While self as context fosters Psychological Flexibility by helping individuals see themselves as separate from their internal experiences self as content limits Psychological Flexibility and heightens distress [21,47].

Similarly, defusion refers to the process of detaching or distancing oneself from unhelpful thoughts (such as pain), beliefs, or memories, rather than becoming entangled or “fused” with them [21]. When patients are fused with their thoughts, their feelings are considered literal truths and directly influence their behaviour and emotions. It is possible that patients with CPP identify themselves as “illness people” or with their diagnosis and their pain. Consequently, they embrace this label, becoming fused with it. Sundström et al. [48] found similar results when examining PF processes and daily functioning in chronic pain patients. In their study, daily functioning was assessed in areas such as work, household chores, social and leisure activities, and relationships, rather than focusing solely on pain interference. Although our outcomes differed, these results support the view that cognitive defusion has a significant impact on patients’ daily lives.

Finally, values refer to guiding principles that define what is important and meaningful to an individual. They can help people to clarify what they want from their life, how they wish to live, and assist in decision-making. Values are considered a source of motivation, guiding actions and behaviours in their direction, even in the presence of difficult thoughts or emotions [21]. In this sense, values act as a resource that can enhance a patient’s well-being and promote Psychological Flexibility. In the Psychological Inflexibility (PI) subscale, its opposite is Lack of values clarity, that indicates a state where individuals’ actions, decisions, or behaviours are misaligned or disconnected from their values and what they find meaningful in life [47]. This disconnection can reduce patients’ well-being and lead to maladaptive behaviours, such as avoidance—whether in therapy or seeking medical help—and catastrophic thinking. These patterns can, in turn, exacerbate symptoms in the context of chronic pain [24,25].

In our results, all these correlations were negative, indicating that patients with chronic pelvic pain are generally fused with their thoughts, perceive their stress as real, and identify themselves with their fears and, possibly, their pain. These processes diminish awareness of the patients’ goals and values, leading to an increased reliance on negative coping strategies for pain management instead of promoting actions directed toward what is meaningful in their lives [23]. Interestingly, the three processes we found to be associated with pain correspond to each of the three domains in the triflex model (e.g., openness, awareness, engagement) [21]. Notably, these results were not confirmed in the analysis of the PI subscale, possibly due to the small sample size. However, associations were found between Cognitive Fusion and Lack of Values Clarity concerning current pain intensity. This suggests that Psychological Inflexibility may play a role not only in pain interference but also in the perception of pain intensity among patients with chronic pelvic pain. In particular, being disconnected from their values and goals, and the use of dysfunctional coping strategies may exacerbate a patient’s perception of pain, fueling experiential avoidance and, consequently, promoting Psychological Inflexibility. Future studies should replicate these findings to further explore the association between Psychological Flexibility and chronic pelvic pain, and to clarify the nature of this relationship. In their study, Maathz et al. [44] reported a longitudinal association between Psychological Inflexibility and pain interference, suggesting that PI may be a precursor of pain rather than a consequence.

The PI composite score and the subcomponent of self as content were both found to be associated with mental quality of life, supporting the hypothesis that greater Psychological Inflexibility, and a strong identification with inner processes, can affect the psychological distress in patients with chronic pelvic pain (CPP). This relationship was further confirmed through the exploration of the links between Psychological Inflexibility and psychological distress. In fact, Psychological Inflexibility, along with its subcomponents, is associated with anxiety, depression, and stress. In this regard, a recent study by Sundström et al. [45] found similar associations between components of Psychological Flexibility and depression in patients with endometriosis. Therefore, considering the significant association observed between psychological distress and pain interference, we may hypothesize that Psychological Flexibility influences both psychological distress and pain interference. However, the correlational nature of this study does not allow us to establish a causal relationship. It is possible that Psychological Flexibility mediates or moderates the relationship between these variables. Future research aimed at exploring this causality is strongly warranted. Ultimately, it is likely that Psychological Inflexibility negatively affects patients’ mental well-being, which in turn influences their perception of pain, creating a vicious cycle. Future studies should further investigate these relationships to confirm these findings.

Finally, Experiential Avoidance and Lack of Values Clarity were also associated with the physical dimension of the SF-12, indicating that Psychological Flexibility impacts patients’ quality of life by interfering with their daily activities. Examining the specific items of the SF-12 suggests that patients with chronic pelvic pain (CPP) may avoid activities that could exacerbate their pain, which, in turn, negatively affects their quality of life. Since some CPP syndromes (e.g., vulvodynia, vaginism) can be trigged by intercourse, patients often renounce their sexual activities [44,49]. Although this avoidance is demonstrated in the field of sexual functioning, it is possible that many other areas of daily life of CPP patients are compromised. Research on the role of PF in quality of life among individuals with chronic pelvic pain is still scarce. Previous studies have primarily focused on sexual quality of life [50] or health-related quality of life (HRQoL) [51], identifying anxiety and depression as predictors of overall quality of life. However, we did not observe similar associations with these variables in our study.

The presence of negative associations between self as context and defusion suggests that patients with chronic pelvic pain (CPP) are often fused with their thoughts, fears, and the identity tied to their diagnosis. To address this, they should engage in specific training aimed at developing the ability to observe their experiences in a detached manner, focusing on external processes rather than internal ones, while adopting a more open and non-judgmental perspective.

## 5. Limitations

This study has several limitations. First, its correlational nature prevents drawing definitive conclusions or inferring the direction of the relationship between Psychological Flexibility, pain, and quality of life. This challenge is compounded by another limitation: the small sample size, which made it difficult to achieve the minimum sample required by the power analysis. These factors constrained the data analysis strategy, and it is possible that the observed effects are underestimated. Additionally, the recruitment of chronic pelvic pain (CPP) patients posed significant challenges. This is partly due to the difficulties patients face in obtaining a diagnosis and partly due to the frequent psychological comorbidities in this population, which may reduce their motivation to seek medical care and participate in research. Finally, our analysis did not allow us to control for the use of pharmacological treatments for CPP, due to the wide heterogeneity of medications reported by participants and the presence of multiple comorbidities with chronic illnesses, which further complicate treatment approaches. Similarly, we were unable to control for pain duration, as patients with CPP often face significant delays in receiving a diagnosis (estimated to average around 10 years). As a result, most participants in our sample reported similar durations of pain, limiting variability in this factor and reducing its utility in the analyses.

## 6. Future Directions

The findings from this study have several important implications for both clinical practice and future research. Psychological Flexibility appears to significantly impact pain perception, pain interference, and quality of life in the delicate population of CPP patients. Therefore, future longitudinal studies are needed to confirm these associations in larger samples, to enhance statistical power and obtain higher-quality data to support our results. In particular, the use of instruments like the MPFI should be pursued to measure all the specific processes of the Psychological Flexibility model. Future research should also address the male population suffering from CPP, which has been largely overlooked in the current literature.

In terms of clinical implications, Psychological Flexibility could serve as a valuable tool to bridge the gap in existing treatments for CPP. Current guidelines [1] on CPP management require a multifactorial approach due to the complexity of the condition. Effective treatment should integrate physical therapy, medication, and psychological support, tailored to the specific needs of each patient. ACT-based interventions, aimed at enhancing Psychological Flexibility, can constitute a valid ally in this population. These interventions have already demonstrated efficacy in other chronic pain conditions, and a single-case study by Chisari et al. [52] yielded encouraging results, showing improvements in pain and sexual function among women with vulvodynia.

Moreover, ACT-based treatments can be individualized, delivered either online or in person, and offered in both individual and group settings. These flexible formats offer economic and logistical advantages for patients, healthcare professionals, and healthcare systems alike.

## 7. Conclusions

In summary, this correlational study explored the association between Psychological Flexibility, Psychological Inflexibility, their subcomponents, psychological distress, pain, and quality of life in women with chronic pelvic pain. The findings suggest that Psychological Flexibility plays a significant role in how pain interferes with daily life, which, in turn, affects both mental and physical quality of life. While the study is limited by certain limitations, it provides valuable insights into the potential role of Psychological Flexibility in pelvic pain management and highlights the promising application of ACT-based interventions as part of an integrated approach to treating these patients.

## Data Availability

The data presented in this study are available on request from the corresponding author.

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
