# Peer review of "The Role of Psychological Flexibility and Psychological Factors in Chronic Pelvic Pain Among Women: A Correlational Study"

_healthcare, 2025, doi:10.3390/healthcare13141697_

Round 1

Reviewer 1 Report

Comments and Suggestions for Authors

Dear authors, congratulations for your hard work. Chronic pelvic pain is a pretty much neglected subject, with great impact on various aspects of a patient's life (social, sexual etc). Your study tries to elucidate the role of psychological factors in females with CPP.

If I may I would like to share some thoughts, comments, with you that you might find helpful.

  1. Your main limitation, as you have already mentioned is your sample, which is in fact small. therefore you cannot have high quality data.
  2. I don't think that you should generalize your findings and correlate them with males also, therefore I don't think thta this is one of your limitations.
  3. I didn't find attached your tables nor the supplementary matterials.
  4. I would like you to share more data regarding the clinics were you recruited women from. You characterize them as specialized Were they public, private only, in what percentage?
  5. Were the questionnaires and scales validated in Italian?
  6. What was the final form of your questionnaire?

Keep up the good work!

Author Response

Dear reviewer,
thank you very much for appreciating our work and for the valuable suggestions.
Please find below our point-by-point response to each of your concerns:

  1. Your main limitation, as you have already mentioned is your sample, which is in fact small. therefore you cannot have high quality data
    We agree that the small sample size limits the statistical power of the study and, consequently, the generalizability of the findings. We have highlighted this limitation in the manuscript, both in the limitations and discussion sections. Nevertheless, considering the clinical sample, we believe our results offer useful preliminary insights and could serve as basis for future studies with larger samples.
  2. I don't think that you should generalize your findings and correlate them with males also, therefore I don't think thta this is one of your limitations
    Thank you for this clarification. We acknowledge your point and have removed the mention of generalizing the findings to male populations from the limitations section. Our sample included only women, and the conclusions are now more appropriately framed within this population.
  3. I didn't find attached your tables nor the supplementary matterials.
    We have now ensured that all tables and the supplementary materials (including the questionnaire and informed consent form) are properly attached to the submission. Please let us know if any materials are still missing
  4. I would like you to share more data regarding the clinics were you recruited women from. You characterize them as specialized Were they public, private only, in what percentage?
    We have added more detail about the clinical settings in the Methods section. The women were recruited from two private specialized gynecological centers in northern Italy, with the support of active patient associations for CPP. This information has been specified in the revised manuscript.
  5. Were the questionnaires and scales validated in Italian?
    Yes, all questionnaires and scales used in the study were validated in Italian. We have added the appropriate references and clarifications in the Measures section to make this explicit.
  6. What was the final form of your questionnaire?
    The final version of the questionnaire included standardized and validated measures only, administered in a fixed order using an online platform (Qualtrics). The final version of the questionnaire is attached at the end of the revised manuscript.

We are grateful for your detailed feedback, which helped us strengthen the manuscript. Please see the attachment to find all the revisions highlighted.

Kind regards,
Chiara Manna
On behalf of all co-authors

Reviewer 2 Report

Comments and Suggestions for Authors

Thank you for paper concerning women’s health. I have some comments:

In introduction, the study should mention the role of ultrasound in excluding the structured pathologies, especially, malignant pelvic diseases in abnormal bleeding women. Please add this point. doi: 10.1007/s40477-022-00732-w

A study flowchart with number of inclusion/exclusion cases should be provided.

Sample size of study was not calculated.

I concern if query tool relating chronic pelvic pain were all validated in population study.

The table results were missed in the main text. No table were present. Thus, the result is not well observed.

Author Response

Dear Reviewer,

Thank you for your thoughtful comments and valuable suggestions. Please find below our point-by-point response to each of your concerns:

  1. In introduction, the study should mention the role of ultrasound in excluding the structured pathologies, especially, malignant pelvic diseases in abnormal bleeding women. Please add this point. doi: 10.1007/s40477-022-00732-w
    Thank you for this insightful suggestion. As recommended, we have revised the Introduction to highlight the role of ultrasound in identifying structural pathologies and excluding malignant conditions in women with CPP. The suggested reference has been appropriately cited.
  2.  A study flowchart with number of inclusion/exclusion cases should be provided.
    We appreciate your comment. However, due to the nature of our cross-sectional correlational design and the single time-point assessment, a flowchart is not applicable. All participants completed the questionnaires, and no dropouts or exclusions occurred post-enrollment. Prior to participation, all individuals were screened by physicians and obstetricians to ensure inclusion criteria were met.
  3. Sample size of study was not calculated.
    Thank you for raising this point. We confirm that a sample size calculation was performed, and the details are reported in paragraph 2.3 of the Methods section.
  4. I concern if query tool relating chronic pelvic pain were all validated in population study
    Thank you for your feedback. All the instruments used have been validated in the Italian population and are widely used in both clinical and research contexts. We have added the appropriate references and clarifications in the Measures section.
  5. The table results were missed in the main text. No table were present. Thus, the result is not well observed
    We apologize for the omission. All result tables have now been correctly attached at the end of the revised manuscript. Each table is clearly labeled and referenced within the Results section to ensure clarity and ease of interpretation.

Please see the attachment to find the revised manuscript.

We thank you again for your constructive feedback, which helped us to improve the quality of our work.

Sincerely,
Chiara Manna
On behalf of all co-authors

Reviewer 3 Report

Comments and Suggestions for Authors

The paper titled makes several original and relevant contributions to a specific gap in the field of chronic pelvic pain (CPP) research. Based on the article, here are specific methodological improvements and additional controls the authors should consider:

  1. The authors acknowledge that the sample size (N=114) was insufficient to achieve the a priori calculated statistical power (119 participants), limiting the use of parametric tests. Future studies should recruit a larger sample to increase statistical power.
  2. No data were reported on pharmacological or psychological treatments participants were undergoing, which could impact pain perception and psychological outcomes. This is an important covariate to control.
  3. While duration of CPP was reported (>1 year for most), it was not controlled for in analysis. Pain chronicity may interact with psychological flexibility and outcomes and should be included as a covariate.

To improve rigor, the authors should increase sample size, stratify by diagnosis, adopt a longitudinal design, and include both clinical and psychosocial controls.

Author Response

Dear reviewer,

thank you for your time, thoughtful comments, and valuable suggestions to improve our manuscript. Please find below our point-by-point responses to your comments. Please see the attachment to find all the highlighted revisions.

  1. The authors acknowledge that the sample size (N=114) was insufficient to achieve the a priori calculated statistical power (119 participants), limiting the use of parametric tests. Future studies should recruit a larger sample to increase statistical power.
    Thank you for this observation. As noted, we acknowledge in the revised manuscript that our sample size did not meet the a priori power calculation (N=119). This limitation is now clearly stated in the Limitations and Discussion sections, along with a recommendation for future studies to include larger samples to ensure sufficient statistical power.
  2. No data were reported on pharmacological or psychological treatments participants were undergoing, which could impact pain perception and psychological outcomes. This is an important covariate to control
    Thank you for your important comment. The use of medications was investigated through an open-ended question included in the Brief Pain Inventory (BPI), specifically item 7. However, the range and types of medications reported by participants were highly heterogeneous due to comorbid conditions often associated with chronic pelvic pain (CPP), such as other chronic illnesses and gynecological disorders. This heterogeneity makes it difficult to statistically control for medication use as a single, unified variable in the analysis. We acknowledge this as a limitation of the study and have clarified this aspect in the revised manuscript.
  3. While duration of CPP was reported (>1 year for most), it was not controlled for in analysis. Pain chronicity may interact with psychological flexibility and outcomes and should be included as a covariate
    We really appreciate this insightful suggestion. While we agree that pain chronicity may affect psychological outcomes and flexibility, our sample was highly homogeneous in this regard, with the majority of participants reporting CPP for more than one year. This limited variance did not allow for meaningful statistical control of this variable as a covariate. We have added this explanation as a limitation in the revised manuscript, emphasizing the need for future studies to consider pain duration more systematically.

Please let us know if further clarification or edits are needed. Thank you once again for your valuable feedback.

Sincerely,
Chiara Manna
On behalf of all co-authors

Round 2

Reviewer 1 Report

Comments and Suggestions for Authors

Dear authors, thank you for your revised manuscript. Best of luck upon your future projects!

Reviewer 2 Report

Comments and Suggestions for Authors

Thank you for your responses.